# Cognitively Inspired Reflective Evolution: Interactive Multi-Turn LLM–EA Synthesis of Heuristics for Combinatorial Optimization

## Abstract

Designing effective heuristics for NP-hard combinatorial optimization problems remains a challenging, expertise-driven task. Recent uses of large language models (LLMs) primarily rely on one-shot code synthesis, producing fragile, unvalidated heuristics and under-utilizing LLMs' capacity for iterative reasoning and structured reflection. In this paper, we introduce **Cognitively Inspired Reflective Evolution (CIRE)**, a hybrid framework that embeds LLMs as interactive, multi-turn reasoners within an evolutionary algorithm (EA). CIRE (i) constructs performance-profile clusters of candidate heuristics to give the LLM compact, behaviorally coherent context; (ii) engages the model in multi-turn, feedback-driven reflection tasks that produce explainable performance analyses and targeted heuristic refinements to broaden the exploration–exploitation frontier; and (iii) integrates and selectively validates these proposals via an EA meta-controller that adaptively balances search. Extensive experiments on benchmark combinatorial optimization show that CIRE yields heuristics that are both more robust and more diverse, achieving consistent, statistically significant gains over one-shot LLM generation, genetic programming baselines, and population-based EAs without LLM feedback. These findings suggest that interactive, cognitively inspired multi-turn reasoning is a promising paradigm for automated heuristic design.

## 1 Introduction

Combinatorial optimization problems (COPs) such as the Traveling Salesman Problem (TSP), vehicle routing, and task scheduling lie at the heart of logistics, network design, and industrial planning. These problems are NP-hard, and exact algorithms become impractical as instance size grows. As a result, practical success relies on heuristics that can produce high-quality approximate solutions under limited computational budgets Blum & Roli (2003). The ability to design strong heuristics is therefore a cornerstone of progress in both operations research and artificial intelligence, with direct implications for real-world decision-making systems.

Despite their importance, heuristics are typically the product of painstaking, expert-driven trial-and-error Burke et al. (2013). The design process requires deep domain knowledge and extensive experimentation, often yielding solutions that are brittle or highly problem-specific. Efforts to automate this process through evolutionary algorithms (EAs) and genetic programming have shown promise Branke et al. (2016), but these methods frequently generate redundant or fragile rules, and struggle to capture higher-level insight into why certain heuristics succeed or fail. Thus, existing approaches can generate candidate heuristics, but they lack systematic mechanisms for *critiquing and refining* those heuristics to boost performance.

The emergence of large language models (LLMs) offers a new opportunity to revisit this challenge Liu et al. (2024). Modern LLMs are not only capable of producing executable code, but also of generating natural-language explanations, comparative analyses, and step-by-step reasoning Wei et al. (2022); Ye et al. (2024); Surina et al. (2025). Recent attempts to apply LLMs for heuristic synthesis generally focus on one-shot code generation, where the model outputs a heuristic implementation that is then evaluated. While attractive, this paradigm often results in unstable or unvalidated solutions, and underutilizes the LLM's potential for iterative reflection and improvement. For example,

AlphaCode uses large-scale sampling and filtering of many candidate programs rather than relying on a single synthesized solution Li et al. (2022). Treating LLMs as static code generators thus fails to exploit their deeper cognitive capabilities—namely, the ability to analyze feedback and self-improve over multiple interactions.

To address this gap, we introduce **Cognitively Inspired Reflective Evolution** (CIRE), a hybrid LLM–EA framework that reconceptualizes LLMs as *interactive, multi-turn reasoners* for automated heuristic design, rather than passive one-shot coders. Instead of producing heuristics in isolation, our approach leverages the LLM's capacity for critique, reflection, and refinement in tandem with evolutionary search. Concretely, CIRE (i) constructs *performance-profile clusters* of heuristics, grouping candidate solutions into behaviorally coherent sets that provide the model with structured context; (ii) engages the LLM in *multi-turn, feedback-driven reflection*, where the model analyzes the strengths and weaknesses of these clusters and proposes targeted refinements that systematically expand the exploration–exploitation space; and (iii) integrates these refinements into an EA meta-controller that adaptively balances exploration and exploitation via selective validation and survival. This synergy between evolutionary search and cognitively inspired reflection transforms heuristic design into an iterative, guided exploration process.

Our contributions are summarized as follows:

- We propose a cognitively inspired LLM–EA framework that redefines LLMs as interactive, multi-turn reasoners within evolutionary search, rather than static code generators.
- We develop a clustering-based reflective feedback mechanism that structures the LLM's analysis around behaviorally coherent groups of heuristics, enabling more informative and generalizable refinements.
- We design a feedback-driven, multi-turn prompting strategy that broadens the exploration–exploitation frontier and integrate it into an EA meta-controller for stable, verifiable search.
- We empirically evaluate our framework on benchmark combinatorial optimization problem, showing that CIRE yields heuristics that are more robust and diverse, with statistically significant gains over one-shot LLM generation, classical genetic programming baselines, and EA search without LLM feedback.

## 2 RELATED WORKS

Research on designing heuristics for NP-hard combinatorial optimization problems—such as the Bin Packing Problem (BPP) and the Traveling Salesman Problem (TSP)—has a long and influential history. Classical heuristics for TSP date back to nearest-neighbor and insertion procedures analyzed by Rosenkrantz et al. (1977), followed by powerful local-improvement strategies such as 2-opt, 3-opt, and the Lin–Kernighan method. In parallel, the BPP has been shaped by simple greedy rules, including First Fit and Best Fit Sgall (2014). These manually crafted heuristics are valued for their efficiency and interpretability, yet they remain brittle, highly problem-specific, and difficult to generalize across distributions Blum & Roli (2003); Burke et al. (2013). Such limitations motivated early attempts to automate the discovery of heuristics. A prominent direction was the development of hyper-heuristics and evolutionary approaches that evolve heuristic rules from primitive components Branke et al. (2016). These methods demonstrated the feasibility of automatic heuristic construction, often surpassing hand-crafted baselines. However, they frequently converged to redundant or fragile structures and produced heuristics that were difficult to interpret or refine. Reinforcement learning and neural combinatorial optimization later extended automated design by training policies directly on optimization tasks, but typically required extensive training data, heavy computation, and suffered from poor out-of-distribution generalization.

A significant leap in AI-driven algorithm design was made by DeepMind's *AlphaCode* Li et al. (2022), which demonstrated that large Transformer models, combined with extensive sampling and filtering, can generate solutions at a competitive programming level. Despite its success, AlphaCode relied heavily on scale and lacked a mechanism for iterative self-improvement—an ability essential for heuristic design, where refinement and error correction are crucial. This limitation motivated a shift toward using LLMs as reasoning agents that iteratively generate, critique, and refine heuristics. FunSearch Romera-Paredes et al. (2023) first showed that evolving LLM-generated programs can

surpass classical heuristics for online bin packing. The Evolution of Heuristics (EoH) framework Liu et al. (2024) extended this idea by co-evolving both code and natural-language "thoughts," improving robustness through reasoning-guided evolution. ReEvo Ye et al. (2024) introduced reflective critique, but its mechanism primarily compared heuristics in pairs, which can limit the richness of feedback and restrict exploration diversity. HSEvo Dat et al. (2025) further addressed diversity loss, and Hemberg et al. Hemberg et al. (2024) integrated LLMs directly as mutation operators within genetic programming pipelines.

A key insight motivating our work arises from recent reinforcement learning studies comparing Direct Preference Optimization (DPO) with Group Relative Policy Optimization (GRPO) Du et al. (2025); Shao et al. (2024). These studies report that group-based preference signals—where the model reasons over sets of trajectories rather than isolated pairs—yield more stable learning dynamics and stronger performance on tasks involving program synthesis and code optimization. This trend highlights a broader principle: group-level feedback provides richer comparative structure, enabling the model to extract more nuanced patterns than pairwise comparisons allow. Inspired by this, heuristic design should similarly move beyond pairwise reflection. While ReEvo's pairwise critique encourages refinement, it restricts the LLM's ability to understand broader behavioral patterns across the population of heuristics. In contrast, group-based reflection enables the LLM to analyze clusters of heuristics, identifying shared failure modes, contrasting exploration strategies, and synthesizing improvements that leverage strengths from multiple groups. This mirrors the advantages of GRPO-like group reasoning: better structural feedback, more stable refinement, and improved coverage of the exploration–exploitation landscape. Overall, the trajectory of prior work—from expert-crafted heuristics, to evolutionary and learning-based automation, to reflective LLM-driven synthesis—reveals persistent challenges: one-shot brittleness, limited interpretability, and insufficient mechanisms for structured improvement. Our framework builds on these insights by combining multi-turn reflection with cluster-based, group-level comparative feedback, enabling LLMs not only to generate heuristics but also to reason across populations of candidates and evolve more generalizable, robust strategies over time.

## 3 METHODOLOGY

### 3.1 OVERVIEW

Our proposed method, **Cognitively Inspired Reflective Evolution (CIRE)**, as shown in **Figure 1**, is motivated by the way human cognition develops strategies through incremental reasoning and self-reflection. Humans rarely arrive at effective solutions in a single attempt; instead, they progress step by step, comparing alternatives, grouping related ideas, reflecting on their strengths and weaknesses, and refining them through iterative deliberation. Inspired by this process, CIRE establishes a multi-turn mechanism that enables large language models (LLMs) to evolve heuristics dynamically, transcending the limitations of static, one-shot generation.

The framework unfolds through stages:

- **Grouping and Behavioral Clustering:** The initial pool of heuristics is organized into groups using two complementary strategies: one ensures the existence of groups with structural or performance similarity, while the other enforces diversity in semantics and design. This dual mechanism establishes a rich basis for reflection, ensuring that feedback is contextualized both within coherent heuristic families and across contrasting perspectives.

- **Reflective Multi-turn Refinement:** Within each group, the LLM engages in a structured multi-turn dialogue. Instead of isolated prompt calls, the model critiques why certain heuristics succeed or fail, extracts comparative insights, and synthesizes refined or entirely novel variants. This reflective evolution mirrors human cognitive processes of critique and incremental improvement, allowing the search to progressively deepen.

### 3.2 GROUPING AND BEHAVIORAL CLUSTERING

**Representation.** Each heuristic candidate is evaluated on a benchmark set $\mathcal{I} = \{1, \ldots, m\}$. Let $e_i(h)$ denote the objective value obtained by heuristic $h$ on instance $i$. To support more ex-

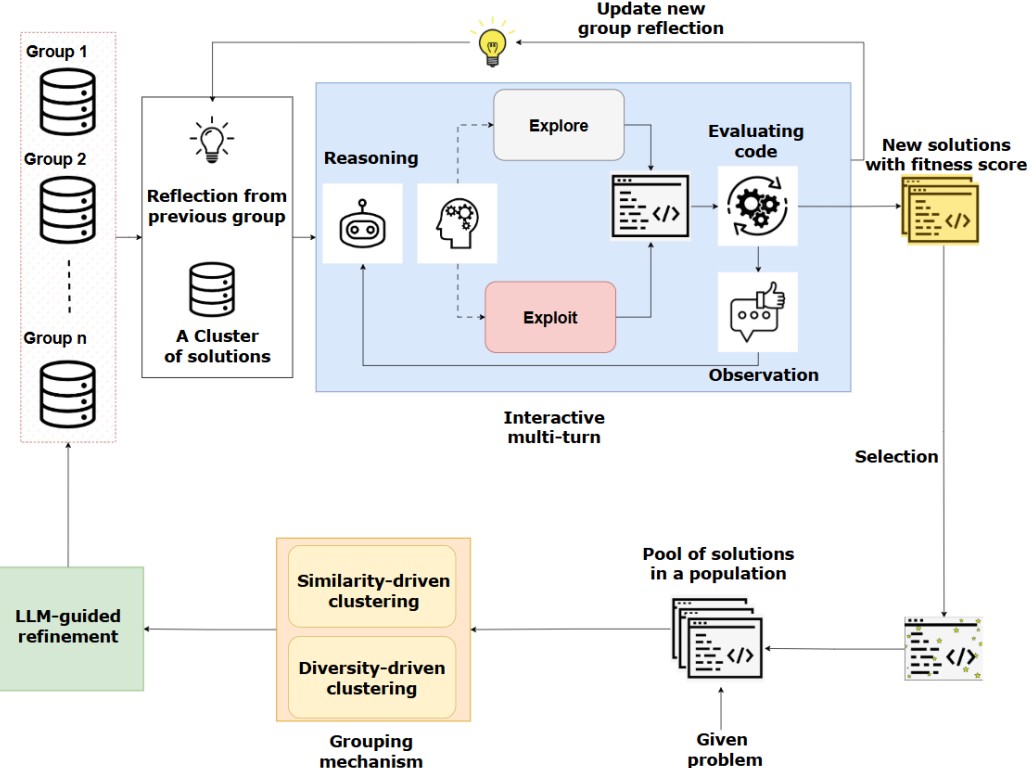

Figure 1: **Cognitively Inspired Reflective Evolution (CIRE):** The framework begins with Population Initialization to create a diverse set of candidate heuristics. Candidates are then organized in Grouping to ensure both coherence and diversity. Reflective Multi-turn Refinement iteratively evaluates, diagnoses, and improves heuristics. Finally, Population Management retains candidates that balance quality and diversity, enabling sustained heuristic evolution.

pressive feedback and nuanced comparison, we encode each heuristic using a normalized performance–profile vector:

$$\mathbf{z}(h) = \frac{\mathbf{e}(h) - \mathbf{e}^*}{\mathbf{e}^*}, \tag{1}$$

where

$$\mathbf{e}(h) = \big(e_1(h), \ldots, e_m(h)\big)^\top, \qquad \mathbf{e}^* = \big(e_1^*, \ldots, e_m^*\big)^\top, \tag{2}$$

and

$$e_i^* = \min_{h' \in \mathcal{H}} e_i(h'), \qquad i = 1, \ldots, m. \tag{3}$$

Here, $e_i^*$ denotes the best-known cost on instance $i$ across all heuristics in $\mathcal{H}$. This instance-wise normalization preserves each heuristic's full performance profile, enabling precise diagnostics, clearer comparisons, and more informative signals for ranking or refinement.

**General idea.** Adequate reflection arises only when the LLM interacts with *well-structured heuristic sets*. Homogeneous groups support fine-grained, instance-level comparison, whereas heterogeneous groups broaden the abstraction space and stimulate creative synthesis. CIRE exploits both perspectives: it first constructs **homogeneous groups** using a static clustering method followed by LLM-based refinement, ensuring internally coherent sets of heuristics. It then forms **heterogeneous groups** by combining heuristics from distinct homogeneous groups, thereby increasing diversity and expanding the reflective space to elicit more innovative heuristic improvements.

HOMOGENEOUS GROUPS (SIMILARITY-DRIVEN)

Homogeneous groups comprise heuristics with similar behavior or structural traits. We quantify similarity between heuristics $h_i$ and $h_j$ by integrating two complementary signals:

- **Performance similarity.** Behavioral similarity is measured in the normalized performance space $\mathbf{z}(\cdot)$ via cosine similarity:

$$\text{sim}_{\text{perf}}(h_i, h_j) \;=\; \frac{\mathbf{z}(h_i)^\top \mathbf{z}(h_j)}{\|\mathbf{z}(h_i)\|_2\,\|\mathbf{z}(h_j)\|_2}, \tag{4}$$

- **Semantic similarity.** Structural resemblance is assessed with CodeBLEU, which extends BLEU by incorporating lexical, syntactic, semantic, and data-flow aspects:

$$\text{sim}_{\text{code}}(h_i, h_j) \;=\; \text{CodeBLEU}(h_i, h_j) \;\in [0, 1]. \tag{5}$$

The overall similarity is defined as a weighted combination of the two signals:

$$\text{sim}(h_i, h_j) \;=\; \alpha \times \text{sim}_{\text{perf}}(h_i, h_j) \;+\; \beta \times \text{sim}_{\text{code}}(h_i, h_j), \qquad \alpha, \beta \geq 0. \tag{6}$$

**Clustering procedure:**

To construct homogeneous groups we represent heuristics as a weighted similarity graph

$$G = (H, E, W), \qquad W_{ij} = \text{sim}(h_i, h_j) \quad ((h_i, h_j) \in E), \tag{7}$$

where $H$ is the set of heuristics and $W \in \mathbb{R}^{|H| \times |H|}$ stores pairwise similarities. Normalize $W$ by its global maximum and form dissimilarities

$$\tilde{W} \;=\; \frac{W}{\max_{p,q} W_{pq}}, \qquad d_{ij} \;=\; 1 - \tilde{W}_{ij}, \qquad D = (d_{ij})_{i,j}. \tag{8}$$

Clustering is sensitive to granularity: too few groups collapses distinctions, while too many fragments structure. CIRE therefore uses a two-phase routine:

1. **Initial over-partitioning.** Apply agglomerative clustering with linkage $L$ to obtain a fine partition

$$(C_1, \ldots, C_m) = \text{Agglomerative}_L(D;\; m), \qquad m \gg 1, \tag{9}$$

   chosen so that clusters are tight:

$$\text{diam}(C_\ell) = \max_{x,y \in C_\ell} d(x, y) \leq \delta, \qquad \delta \ll 1. \tag{10}$$

2. **LLM-guided refinement.** Provide the full over-partition $\mathcal{C}_0 = \{C_1, \ldots, C_m\}$ to an LLM which returns a globally restructured partition

$$\mathcal{C}^{\text{ref}} = \Phi_{\text{LLM}}(\mathcal{C}_0, W), \tag{11}$$

   subject to the partition constraints

$$\bigcup_{C \in \mathcal{C}^{\text{ref}}} C = H, \qquad C_p \cap C_q = \emptyset \quad (p \neq q). \tag{12}$$

HETEROGENEOUS GROUPS (DIVERSITY-DRIVEN)

**Diversity via entropy.** The diversity of a homogeneous cluster $G = \{h_1, \ldots, h_m\}$ is quantified by the entropy of its internal similarity distribution. Since the composite similarity sim is symmetric, we consider only unordered pairs $(i, j)$ with $i < j$. Normalized affinities are defined as

$$p_{ij} \;=\; \frac{\text{sim}(h_i, h_j)}{\sum_{u < v} \text{sim}(h_u, h_v)}, \qquad i < j, \tag{13}$$

which form a valid probability distribution. The entropy score of $G$ is then

$$\mathcal{H}(G) \;=\; -\sum_{i < j} p_{ij} \, \log p_{ij}, \tag{14}$$

with larger values indicating higher internal heterogeneity.

**Entropy-weighted sampling.** Given $k$ homogeneous clusters $\{G_1, \ldots, G_k\}$, sampling weights are assigned proportionally to their entropy values:

$$w_i = \frac{\mathcal{H}(G_i)}{\sum_{j=1}^{k} \mathcal{H}(G_j)}, \qquad i = 1, \ldots, k. \tag{15}$$

For a target heterogeneous group size $L$, the contribution of each cluster is

$$L_i = \lfloor w_i \cdot L \rfloor, \tag{16}$$

$L_i$ heuristics are drawn uniformly at random from $G_i$. The final heterogeneous group is the union

$$G^{\text{het}} = \bigcup_{i=1}^{k} S_i, \qquad S_i \subseteq G_i, \ |S_i| = L_i. \tag{17}$$

### 3.3 REFLECTIVE MULTI-TURN REFINEMENT

CIRE organizes the search as a sequence of iterative turns, each following the loop

$$\text{observe} \ \rightarrow \ \text{reason} \ \rightarrow \ \text{act}.$$

This cycle ensures that each LLM invocation contributes to a coherent refinement trajectory rather than producing isolated trials.

**State and reflection.** At turn $t$, the LLM is provided with a compact state representation

$$S_t = \{(h_j, \text{diag}_j)_{j=1}^{m}, \ \text{history}_{<t}, \ \text{directions}\} \tag{18}$$

$h_j$ are candidate heuristics in the group and $\text{diag}_j$ are diagnostic features (cost, delta-improvement vector). Reflection over $S_t$ allows the model to identify structural weaknesses, recurring failure modes, and latent strengths, thereby grounding subsequent reasoning in accumulated knowledge.

**Adaptive strategy: exploration vs. exploitation.** Conditioned on the reflective analysis, the model decides between two complementary modes. *Exploration* is triggered when recent refinements plateau or converge to structurally similar outcomes, signaling entrapment in a local optimum. In this mode, the LLM proposes divergent heuristics, such as new operators or recombinations across clusters, to enlarge the search horizon. *Exploitation* is selected when promising candidates are detected. Here, the model performs targeted refinement, including parameter tuning, incremental patching, or structural polishing, to systematically transform partial successes into competitive solutions. This dynamic alternation between diversification and intensification is critical for avoiding stagnation while consolidating gains.

**Action and observation.** Once a strategy is chosen, the model outputs actionable artifacts—such as patches, new DSL entries, or modified code—accompanied by a short rationale and confidence estimate. With probability $p$, additional performance observations relative to the current best are injected into $S_t$, sharpening the reflective analysis and aligning the strategy with outcome-based evidence. A fixed maximum-turn budget further regulates the trajectory: early turns favor exploratory breadth, while later turns naturally prioritize exploitative depth.

**Resulting workflow.** The integration of reflection, adaptive strategy selection, and observation-driven feedback yields coherent refinement trajectories $\{h^{(1)}, h^{(2)}, \ldots, h^{(T)}\}$. This workflow achieves sample-efficient improvement, systematically escaping local optima while progressively optimizing promising directions—outcomes unattainable under naive re-prompting.

## 4 EXPERIMENTAL

### 4.1 EMPIRICAL EVALUATION

For evaluation, we seek to answer the following question:

**RQ1.[Effectiveness]** How effectively does CIRE generate higher-quality solver code compared to baseline approaches and the effect across different LLM types?

**RQ2.[Reasoning]** How does CIRE behave across successive reasoning iterations, and how do these iterations influence code correctness and solution quality?

**RQ3.[Ablation Study]** How critical is each component of our method to overall performance?

**Datasets.** We evaluate CIRE on two canonical combinatorial optimization tasks: TSP and Online Bin Packing. The TSP benchmark spans instance sizes from 10 to 200 nodes, while the Bin Packing benchmark covers capacity settings from 100 to 500.

**Models.** We employ DeepSeek V3, a state-of-the-art large language model (LLM), as the backbone for heuristic generation.

**Baselines.** We benchmark against competitive state-of-the-art methods, including EoH (Liu et al. (2024)) and Re-evo (Ye et al. (2024)), as well as several classical heuristic approaches.

**Metrics.** We assess performance using the *optimality gap*, defined as the relative deviation of a solution from the corresponding optimal or best-known value.

## 4.2 RESULTS

**RQ1. [Effectiveness]** We compare our method with strong baselines on BPP and TSP, and evaluate its robustness across different LLM backbones.

Table 1: Online bin packing results. Fraction of excess bins to lower bound (lower is better) on Weibull instances.

| Heuristic | Capacity = 100 | | | Capacity = 300 | | | Capacity = 500 | | |
|---|---|---|---|---|---|---|---|---|---|
| | 1k | 5k | 10k | 1k | 5k | 10k | 1k | 5k | 10k |
| First Fit | 5.32% | 4.40% | 4.44% | 1.34% | 0.93% | 0.92% | 0.25% | 0.50% | 0.50% |
| Best Fit | 4.87% | 4.08% | 4.09% | 1.19% | 0.84% | 0.86% | 0.25% | 0.50% | 0.47% |
| EoH | 3.03% | 2.15% | **0.33%** | 0.60% | 0.63% | 0.58% | 0.25% | 0.50% | 0.47% |
| ReEvo | 3.78% | **0.80%** | **0.33%** | 1.04% | 0.27% | 0.19% | 0.25% | 0.50% | 0.47% |
| CIRE (ours) | **2.34%** | 1.13% | 0.59% | **0.30%** | **0.24%** | **0.16%** | **0.25%** | **0.50%** | **0.45%** |

Table 2: Baseline OR-Tools Results for the Traveling Salesman Problem.

| Heuristic | TSP10 (%) | TSP20 (%) | TSP50 (%) | TSP100 (%) | TSP200 (%) |
|---|---|---|---|---|---|
| EoH | 3.52 | 9.33 | 10.24 | 11.39 | 15.58 |
| ReEvo | 4.22 | 6.74 | 11.63 | **11.01** | 15.58 |
| CIRE(ours) | **2.11** | **6.74** | **9.20** | 12.64 | **11.46** |

**Quantitative Evaluation.** Across all bin-packing settings (Table 1), **CIRE** consistently outperforms both classical heuristics and recent adaptive methods. Under tight capacity ($C$=100), CIRE reduces the excess fraction to **2–3%**, compared to **4–5%** for First Fit and Best Fit. At medium capacity ($C$=300), CIRE achieves below **0.3%** excess on long streams, while adaptive baselines such as ReEvo fluctuate between **0.2–0.6%**. Even at large capacities ($C$=500), where all methods converge, CIRE maintains a measurable advantage. A similar trend appears in TSP benchmarks (Table 2), where CIRE achieves consistently lower optimality gaps than state-of-the-art LLM-based approaches (EOH, ReEvo) across instance sizes $n \in [10, 200]$. These results highlight CIRE's ability to generalize across problem scales and combinatorial structures.

**Effect of LLM Backbone.** To isolate whether performance stems from the LLM or from the CIRE workflow itself, we evaluate CIRE with a diverse set of models—`deepseek-v3-0324`, `kimi-k2-instruct`, `qwen3-coder`, `480b-35a`, and `glm-4.5` which shown in Table 3. (These models span instruction-tuned, code-centric, and large general-purpose LLM families, providing a representative capability spectrum.) Across all backbones, CIRE remains highly stable: the best average gap is obtained with `deepseek-v3-0324` (**1.13**) and the worst with

Table 3: Online Bin Packing: Method Across Models

| Heuristic | Model | Gap(%) |
|---|---|---|
| CIRE | deepseek-v3-0324 | 1.13 |
| CIRE | kimi-k2-instruct | 1.95 |
| CIRE | qwen3-coder-480b-35a | 1.41 |
| CIRE | glm-4.5 | 1.20 |
| EOH | deepseek-v3-0324 | 2.15 |

Table 4: Coefficient Tuning for TSP 50

| $\alpha$ | Gap(%) |
|---|---|
| 0.2 | 15.79 |
| 0.4 | 14.06 |
| 0.5 | **9.20** |
| 0.6 | 15.97 |
| 0.8 | 12.5 |

Table 5: Analysis of Reasoning Categories, their frequency, and examples.

| Reasoning Category | Description | Freq | Example |
|---|---|---|---|
| Paradigm shift | Completely changes the algorithm family. | 16 | Given that the highest-performing known approach is "Stabilized Harmonic-Arctanh" ... I should explore this proven algorithmic family rather than tuning the existing simpler approaches. |
| Heuristic modification | Same algorithm family and pipeline structure, but decision/scoring logic is substantially rewritten. | 89 | ... Given the significant performance gap and clear indication that Worst-Fit works better, we should focus on refining this approach ... . |
| Hyperparameter tuning | Only numeric changes (weights, thresholds, constants) with formulas/pipeline unchanged. | 75 | ... Given the regression, we should: 1. Revert to the simpler 0.0386 version 2. Make minimal adjustments to core parameters ... . |

`kimi-k2-instruct` (**1.95**). The narrow performance band indicates that strong results arise *not* from the raw power of the underlying LLM, but from CIRE's structured reasoning and refinement workflow. This independence from a specific backbone underscores CIRE as a robust and general optimization framework.

**RQ2.[Reasoning]** We investigate how CIRE's multi-turn reasoning shapes code correctness and solution quality. Using LLM-assisted classification, each refinement step is categorized as a *paradigm shift*, *heuristic modification*, or *hyperparameter tuning* (Table 5). The distribution shows a dominant reliance on heuristic modification (**89**) and hyperparameter tuning (**75**), with paradigm shifts occurring only rarely (**16**), indicating that CIRE quickly commits to exploitation after brief initial exploration.

**Turn-by-turn analysis** (Figure 2) further reveals a consistent pattern: exploration via paradigm shifts occurs mainly at the first and final iterations. In contrast, the middle iterations concentrate on fine-grained exploitation. This behavior highlights a key insight: LLM-based optimization naturally follows an *explore-then-exploit* reasoning trajectory, with focused mid-stage refinement driving most of the performance gains.

**RQ3.[Ablation Study]** To disentangle the contribution of each component in our method, we conduct a comprehensive ablation study.

**Effect of Multi-turn Refinement and Solution Grouping.** Table 6 reports the results on the TSP50 benchmark. The full model achieves a **9.20%** optimality gap, while ablating *either* multi-turn refinement or solution grouping degrades performance to **17.18%**, the worst gap among all variants. This consistent degradation highlights that our strong performance cannot be attributed to a single

Table 6: Ablation study analyzing the impact of multi-turn and grouping mechanisms.

| Setting | Gap(%) |
|---|---|
| w/o multi-turn | 17.18 |
| w/o grouping (randomly choose group) | 15.97 |
| **Ours (Full method)** | **9.20** |

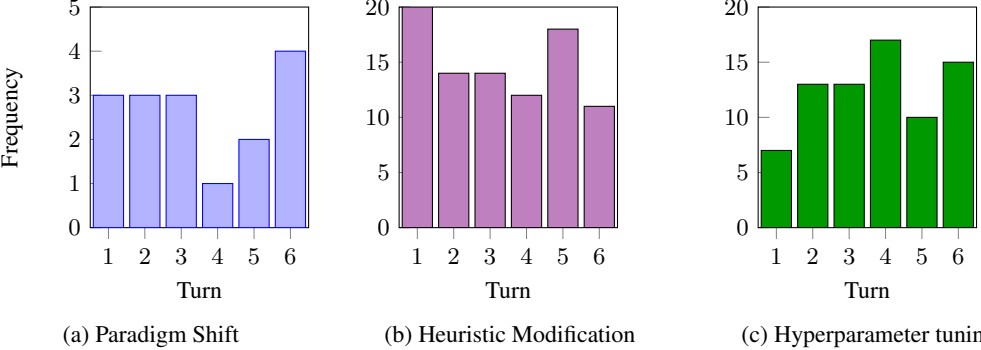

(a) Paradigm Shift     (b) Heuristic Modification     (c) Hyperparameter tuning

Figure 2: Frequency distributions of turns across three different scenarios.

design choice. Instead, it is the *joint effect* of multi-turn reasoning and grouping—both essential for stabilizing the LLM's code generation and guiding it toward higher-quality optimization solutions.

**Effect of Similarity Coefficient Tuning.** We further investigate the impact of the parameter $\alpha$, which balances behavioral similarity and CodeBLEU-based semantic similarity in our hybrid similarity metric. As shown in Table 4, the best performance is achieved at $\alpha = 0.5$, indicating that **both behavioral and semantic signals are indispensable**. This balanced configuration yields the most reliable similarity estimates, and we adopt $\alpha = 0.5$ across all experiments.

**Summary of Findings.** Across all ablations, we observe that removing any component—multi-turn refinement, grouping, or balanced similarity estimation—leads to consistent and measurable performance degradation. These results provide strong evidence that our method derives its effectiveness from a carefully designed combination of components, each playing a distinct and complementary role in enabling LLMs to generate high-quality optimization code.

## 5 CONCLUSION

CIRE reconceptualizes LLM-based heuristic discovery as a reflective, multi-turn refinement process in which each model invocation contributes to a coherent trajectory of reasoning rather than an isolated trial. By embedding diagnostic feedback into every turn, the framework establishes a foundation of reflection that allows the model to recognize strengths, diagnose weaknesses, and build on structural patterns uncovered in earlier attempts. This reflective state underpins a principled balance between exploration—introducing qualitatively new strategies to escape local optima and broaden the search space—and exploitation, where targeted tuning systematically enhances promising heuristics by refining parameters, operators, or structural decisions. The integration of observation signals further grounds this process in empirical evidence, aligning model reasoning with measurable progress and preventing divergence into unproductive paths. Through this interplay of reflection, adaptive decision-making, and performance-driven guidance, CIRE achieves both robustness and sample efficiency while advancing a general methodology in which LLMs operate as adaptive problem solvers rather than static generators. This perspective lays a professional foundation for extending multi-turn refinement beyond the studied benchmarks, offering a principled blueprint for deploying LLMs in a wide range of combinatorial optimization domains where iterative reasoning and adaptive search are indispensable.

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

# 6 APPENDIX

## 6.1 PROMPT

---

**Online Bin Packing Prompt Formulation**

```
TASK SUMMARY

You are an AI assistant whose job is to iteratively produce and
    refine Python heuristic implementations for the Bin Packing
    Online Problem.
You will be given an existing heuristic (or helper functions). Use
    multi-turn reasoning: at each turn you must reflect, then
    either **explore** a new heuristic family or **exploit**
    (refine) the last submitted heuristic, and finally receive an
    observation/feedback from the environment.

---

### FUNCTION CONTRACT (must be strictly respected)
- Language: Python only. Only standard library and numpy allowed
    (if already used by provided code).
- Required signature:
    def score(item, bins)
- Input arguments:
    - item: int # size of current item
    - bins : Numpy arrays # the rest capacities of feasible bins,
        which are larger than the item size.
- Return: scores (Numpy array)
- Correctness rules:
    - 'item' is of type int
    - 'bins' and 'scores' are both Numpy arrays.
```

---

**Travelling Salesman Problem Prompt Formulation**

```
TASK SUMMARY

You are an AI assistant whose job is to iteratively produce and
    refine Python heuristic implementations for the Travelling
    salesman problem.
Given a set of nodes with their coordinates, \
you need to find the shortest route that visits each node once and
    returns to the starting node. \
The task can be solved step-by-step by starting from the current
    node and iteratively choosing the next node. \
You will be given an existing heuristic, Let use multi-turn
    reasoning: at each turn you must reflect, then either
    **explore** a new heuristic family or **exploit** the last
    submitted heuristic, and finally receive an
    observation/feedback from the environment.

### FUNCTION CONTRACT (must be strictly respected)
- Language: Python only. Only standard library and numpy allowed
    (if already used by provided code).
- Required signature:
    def select_next_node(current_node, destination_node,
        univisited_nodes, distance_matrix)
- Input arguments: This function should accept 4 inputs:
    'current_node', 'destination_node', 'univisited_nodes',
    'distance_matrix'
```

```
- Output: The function should return 1 output: 'next_node'
- Correctness rules: 'current_node', 'destination_node',
   'next_node', and 'unvisited_nodes' are node IDs.
   'distance_matrix' is the distance matrix of nodes. All are
   Numpy arrays.
Do not give additional explanations.
```

## Prompt Template for the THINK Step

```
ALWAYS REMEMBER THAT, LOWER fitness score = BETTER solution.
First,based on the evaluation result from <observation> or GROUP
   REFLECTION,
you should do some critical reasoning about the previous
   approach(s) ABOUT:
+ its logical algorithm
+ its heuristic components/hyperparamters/features specifically.
   Then, think about the affect of these parameters/hyperparamters
   to the fitness score result (in detailed).

Then, you can:

1. Explore a totally new approach, to make some experiments to get
   informations.
OR
2. Focus on the behaviour of the heuristic features/components
   from the fitness result to tune them and get better result from
   the test evaluation.

You are ONLY allowed to do reasoning, NOT to generate code.
Note that, your reasoning should be very BRIEF but STILL critical
   and concise, focus on analyzing the heuristic
   components/features.
At the last of your response, there must be one of the tags
   <explore> or <exploit>, which indicate your decision.
```

## Prompt Template for the Exploration Phase

```
Now, BASED solely on your REASONING, generate EXACTLY ONE solution
   for exploring.
Your output MUST be exactly the SAME as the following format:
<explore>
<algorithm>
# clear and completete algorithm description of the proposed
   heuristic.
</algorithm>

# the completely new Python function implementation for the
   algorithm in <algorithm> : `score(...)` (only code inside
   ``).

</explore>
OUTPUT RULE:
Always output exactly one <explore> block containing both
   <algorithm> and , nothing else.
```

---

**Prompt Template for the Exploitation Phase**

```
Now, BASED solely on your REASONING, generate EXACTLY ONE solution
    for exploiting.
Your output MUST be exactly the SAME as the following format:
<exploit>
<algorithm>
# Clear algorithm description of the improvements you're making to
    the selected algorithm
</algorithm>

# Complete and concise Python function implementation with your
    refinements: 'score(...)'

</exploit>
OUTPUT RULE:
Always output exactly one <exploit> block containing both
    <algorithm> and , nothing else.
```

## 6.2 REASONING BEHAVIOR

As shown in Fig. 3, the cognitively inspired reflective evolution process demonstrates how score dynamics and strategic adaptation interact to drive progress beyond local optima. At the outset, when the system observed a fitness of $0.042$, considerably worse than the best-known score of $0.031$, it recognized stagnation and initiated an exploratory shift. This led to the generation of Adaptive Resonance Packing (ARP), a structurally novel heuristic that improved the score to $0.036$. Although this gain was modest, the introduction of ARP provided a fertile ground for subsequent refinements.

Building on this foundation, the process transitioned into targeted exploitation. By tuning ARP through dynamic bandwidth control, exponential gap weighting, and softmax normalization, the system sought to consolidate and optimize the idea. While the score did not improve beyond $0.036$, this stage illustrates the reflective nature of the method: rather than abandoning a promising approach, it strategically invested in fine-tuning, ensuring stability before pursuing further change.

When performance remained unchanged, the system's observation mechanism signaled diminishing returns and prompted a more dramatic innovation. This shift produced Quantum Tunneling Bin Packing (QTBP), a probabilistic mechanism inspired by tunneling to bypass local barriers. Crucially, this step reduced the score to $0.023$, surpassing the best-known baseline of $0.031$.

The trajectory of scores and strategies highlights the essence of our approach: exploration introduces qualitatively new directions, exploitation consolidates promising structures, and reflective reasoning determines when a radical innovation is necessary. In contrast to naive retries that repeat search attempts, our method interprets performance signals to decide between exploration and exploitation. This enables it to escape stagnation and uncover breakthroughs otherwise inaccessible to conventional search.

**Cognitively Inspired Reflective Evolution**

Observation: You still can't beat the best fitness score 0.031

**<reasoning>** The current approach achieved 0.042 which is worse than the best known solution... stuck in local optima ... should explore...

**<explore>** Adaptive Resonance Packing (ARP): Novel Approach... Creates "acoustic" base on harmonic relationships...

Score: 0.036

Observation: Your code is better but still can't beat the best fitness...

**<reasoning>** Adaptive Resonance Packing is promising ... Exploit via target tunning ...

**<exploit>** Improved ... with simplifi dynamic bandwidth, exponential gap weighting, softmax normalization...

Score: 0.036

Observation: Your score is unchanged as previous step.

**<reasoning>** Score is unchanged as ... Need more dramatic shift, exploit with broader innovation...

**<exploit>** ...Quantum Tunneling Bin Packing (QTBP) introducing probabilistic tunneling to jump local barriers, blending ...with quantum-inspired scoring.

Score: 0.023

Figure 3: **Qualitative example of reflective reasoning in CIRE:** Observations guide the LLM to alternate between exploration, exploitation, and innovation, resulting in progressive improvement of heuristic quality.

