# OpenReview forum: "Cognitively Inspired Reflective Evolution: Interactive Multi-Turn LLM–EA Synthesis of Heuristics for Combinatorial Optimization"
_ICLR.cc/2026/Conference — ICLR 2026 Conference Withdrawn Submission_

### Official Review · Reviewer_tdXD · 2025-10-29

**Soundness:** 2
**Presentation:** 2
**Contribution:** 2
**Rating:** 4
**Confidence:** 4

**Summary:**

The paper proposes CIRE, a cognitively inspired framework that embeds a large language model as a multi-turn reasoner inside an evolutionary algorithm to automate heuristic design for combinatorial optimization. CIRE clusters candidate heuristics by performance profiles to give the LLM compact, behaviorally coherent context, then elicits reflective analyses and targeted refinements. An EA meta-controller selectively validates proposals to balance exploration and exploitation. On online bin packing, CIRE yields more robust, diverse heuristics and statistically significant gains over one-shot LLM code generation, genetic programming, and EAs without LLM feedback, reducing optimality gaps and excess-bin fractions under tight capacities.

**Strengths:**

All the proposed components are well-motivated. Multi-turn critique/refinement leverages LLM reasoning beyond one-shot code.
Performance-profile clustering provides compact, behaviorally coherent prompts that improve generalizable updates.
EA meta-controller balances exploration/exploitation with selective validation for stability.

**Weaknesses:**

The presentation could be improved. Section 4.2 is somewhat hard to follow.

The fatal weakness is the lack of validation. The authors should evaluate their methods on more datasets and problem types, and with additional LLMs. The experiments also lack an ablation study for each algorithmic component.

**Questions:**

See weaknesses.

---

> ### Author Response · Authors · 2025-11-23
>
> Dear Reviewer tdXD,
>
> We sincerely appreciate your detailed and insightful comments. We have thoroughly analyzed all raised concerns and present our point-by-point responses below.
>
>
> ***W1***
>
> We sincerely thank the reviewer for pointing out the clarity issues in Section 4.2. We have taken this opportunity to *substantially restructure the paper* to improve both readability and logical flow.
>
> Major Revisions:
>
> **- Restructuring**: To streamline the presentation and create space for the comprehensive experiments (requested by Reviewers), we have removed the dedicated "Motivation" section (originally Section 3) and integrated key points into the Introduction.
>
> **- Renumbering & Rewriting**: Consequently, the original Section 4.2 has been moved to Section 3.2 in the revised PDF.
>
> **- Clarity Improvements**: In this newly designated Section 3.2, we have rewritten the content to be more accessible. In particular, we refined the presentation of the *heuristic encoding* to more clearly articulate the underlying idea and the steps of the method, ensuring that the procedure is now easier to understand and follow. We also reformulated the **""General idea"** to present the group-construction strategy (homogeneous clustering followed by LLM refinement and subsequent heterogeneous mixing) as a straightforward, principled workflow; we emphasize that this group-based reflection is both novel and practically effective for eliciting targeted, creative heuristic refinements.  Finally, we performed a paper-wide edit to improve wording, consistency in notation, and paragraph flow, enhancing overall readability.
>
> ***W2***
>
> We took this feedback very seriously and have utilized the rebuttal period to conduct a **major overhaul of our experimental validation**. We believe the new results in the revised PDF fully mitigate this "fatal weakness."
>
>
> **1.** We have expanded our evaluation beyond Online Bin Packing to include the TSP (see more for revised PDF):
>
> | Heuristic | TSP10 (%) | TSP20 (%) | TSP50 (%) | TSP100 (%) | TSP200 (%) |
> |---|---|---|---|---|---|
> | EoH | 3.52 | 9.33 | 10.24 | 11.39 | 15.58 |
> | ReEvo | 4.22 | 6.74 | 11.63 | 11.01 | 15.58|
> | CIRE(ours) | 2.11 | 6.74 | 9.20 | 12.64 | 11.46 |
>
>
> **2.** We verified our method on diverse model families (DeepSeek, Kimi, Qwen, GLM) on the Online Bin Packing dataset (Weibull distribution, 5k items, Capacity 100).
> | Heuristic | Model | Gap |
> |:---------:|:-----:|:---:|
> | CIRE | deepseek-v3-0324 | 1.13 |
> | CIRE | kimi-k2-instruct | 1.95 |
> | CIRE | qwen3-coder-480b-35a | 1.41 |
> | CIRE | glm-4.5 | 1.20 |
>
> Results demonstrate that CIRE is highly stable across diverse model families. The optimality gap remains consistently low (< 2%) across all backbones, ranging from a minimum of **1.13%** (DeepSeek-V3) to a maximum of only **1.95%** (Kimi-k2).This narrow variance demonstrates that performance gains arise from CIRE’s *multi-candidate reasoning and iterative refinement*, rather than any specific model, establishing it as a* robust, model-agnostic framework for combinatorial optimization.*
>
> **3.** We have added a comprehensive ablation study (Table 6)  to isolate the impact of each algorithmic component. (We use TSP50 instances to isolate the effect of each algorithmic element.)
>
> | Setting | Gap|
> |:-------:|:------:|
> | w/o multi-turn | 17.18 |
> | w/o grouping (randomly choose group) | 15.59 |
> | **Ours (Full method)** | **9.20** |
>
> Removing the multi-turn refinement causes the performance gap to nearly double (from 9.20% $\to$ 17.18%). This empirically proves that the iterative feedback loop is the primary driver of our performance.
>
>
> **4.** We further analyze the impact of the similarity coefficient $\alpha$, which balances behavioral and CodeBLEU-based semantic similarity in our hybrid grouping metric (on the TSP50 dataset). As shown below, performance peaks at $\alpha = 0.5$, confirming that both behavioral and semantic signals are essential. This result also demonstrates that the parameter choices in our method are **carefully designed and empirically validated.**
>
>
> | $\alpha$ | Gap(%) |
> |:--------:|:------:|
> | 0.2 | 15.79 |
> | 0.4 | 14.06 |
> | **0.5** | **9.20** |
> | 0.6 | 15.97 |
> | 0.8 | 12.5 |
>
> **5.**  We also analyze CIRE’s reasoning dynamics to understand the source of its strong performance. Using LLM-assisted classification with human verification, each refinement step is categorized as a paradigm shift, heuristic modification, or hyperparameter tuning (Table 5, revised version). CIRE quickly settles into efficient exploitation, dominated by heuristic refinements and parameter adjustments, with paradigm shifts invoked only when structurally necessary. This leads to *high sample efficiency* and avoids unnecessarily long reflection chains. The **turn-by-turn analysis** (Figure 2, revised version) further reveals a clear *explore → exploit* pattern: exploration occurs at the beginning and end, while the middle iterations focus on precise, stability-oriented refinement.

---

> > ### Comment · Reviewer_tdXD · 2025-11-26
> >
> > Thanks for your response. The results are promising. However, this paper requires substantial revision before acceptance at a top-tier conference like ICLR.

---

### Official Review · Reviewer_hmGT · 2025-10-29

**Soundness:** 2
**Presentation:** 2
**Contribution:** 2
**Rating:** 2
**Confidence:** 4

**Summary:**

The paper proposes CIRE, a hybrid framework that embeds an LLM as a multi-turn “reflective” reasoner inside an evolutionary algorithm to synthesize heuristics for combinatorial optimization (e.g., TSP, BPP). CIRE clusters candidate heuristics by performance profiles to give the LLM structured context, prompts the model to analyze strengths/weaknesses and propose targeted refinements, and uses an EA meta-controller to validate and balance exploration vs. exploitation. Experiments report consistent, statistically significant gains over one-shot LLM generation, genetic programming, and EA baselines; qualitatively, CIRE evolves novel strategies (e.g., ARP, QTBP) that surpass prior best scores.

**Strengths:**

Clear, well-motivated shift from one-shot LLM code synthesis to iterative, cognitively inspired reflection.

Thoughtful design with performance-profile clustering + multi-turn feedback + EA meta-control, which addresses key issues in current LLM+EA methods.

Empirical evidence suggests superior robustness/diversity of heuristics on BPP.

**Weaknesses:**

- The paper lacks a comprehensive evaluation on more problems and datasets.
- Quantitative details are light in the excerpted text: dataset sizes, statistical test specifics, runtime/compute budgets, etc. Ablation is lacking. Stronger reporting would aid reproducibility.
- Potential sensitivity to clustering choices isn’t fully dissected; robustness across LLMs and hyperparameters remains to be demonstrated.

**Questions:**

How are the weights α and β in the similarity metric (behavioral vs. CodeBLEU semantic) chosen, and how sensitive are results to them?

What is the stopping criterion for multi-turn refinement?

---

> ### Author Response · Authors · 2025-11-23
>
> Dear Reviewer hmGT,
>
> We greatly appreciate your thoughtful and constructive feedback. Each comment has been carefully considered, and we provide our detailed, point-by-point responses below.
>
> **W1** :
>
> We acknowledge the need for a more rigorous empirical evaluation and have updated the manuscript to include the following:
>
> **Expanded Datasets**:
> We have expanded our evaluation to cover a broader range of problem scales (TSP-10 to TSP-200). As summarized in following table, CIRE achieves superior or highly competitive performance across all scales. Notably, on the most challenging instance (**TSP-200**), CIRE significantly outperforms the strongest baseline (EoH) by **4.12%** (11.46% vs 15.58%), demonstrating robust scalability..
> | Heuristic | TSP10 (%) | TSP20 (%) | TSP50 (%) | TSP100 (%) | TSP200 (%) |
> |---|---|---|---|---|---|
> | EoH | 3.52 | 9.33 | 10.24 | 11.39 | 15.58 |
> | ReEvo | 4.22 | 6.74 | 11.63 | 11.01 | 15.58|
> | CIRE(ours) | 2.11 | 6.74 | 9.20 | 12.64 | 11.46 |
>
> **W2**:
>
> **Ablation Studies & Computational Efficiency Ablation Study**:
>
>
> We have added a comprehensive ablation study conducted on TSP50 instances to isolate the impact of each algorithmic component in the following table.  Results confirm that our grouping strategy and multi-turn interaction are synergistic. Removing the grouping mechanism (random selection) nearly doubles the optimality gap, while removing multi-turn capabilities yields the worst performance (**17.18%**).
> | Setting | Gap|
> |:-------:|:------:|
> | w/o multi-turn | 17.18 |
> | w/o grouping (randomly choose group) | 15.59 |
> | **Ours (Full method)** | **9.20** |
>
> **Compute Budgets**
>
> Benchmarking on DeepSeek-V3 demonstrates that CIRE incurs a marginal average cost of **$0.41/run** (vs. EoH: $0.33, ReEvo: $0.25). This confirms that our grouping mechanism effectively compresses the search space, enabling complex multi-candidate reasoning with minimal token overhead.
>
> **W3**:
>
> We addressed concerns regarding sensitivity and model generalization with the following experiments:
>
> **Clustering Sensitivity**
>
> We analyzed the weight α , which balances behavioral vs. CodeBLEU semantic similarity (where  β = 1- α). The following indicates a clear optimum at α=0.5. Extreme values (ignoring either behavioral or semantic features) degrade performance, validating our hybrid metric.
>
> | α  | Gap(%)|
> |:-------:|:------:|
> | 0.2 | 15.79 |
> | 0.4 | 14.06 |
> | 0.5 | **9.20** |
> | 0.6 | 15.97 |
> | 0.8 | 12.5 |
>
>
> **LLM Generalization**
>
> To demonstrate model agnosticism, we benchmarked CIRE across diverse architectures (DeepSeek, Kimi, Qwen, GLM) on the Online Bin Packing dataset (Weibull, 5k items). Table 4 shows consistent performance (Gap < 2%) across all backbones.]
> | Heuristic | Model | Gap |
> |:---------:|:-----:|:---:|
> | CIRE | deepseek-v3-0324 | 1.13 |
> | CIRE | kimi-k2-instruct | 1.95 |
> | CIRE | qwen3-coder-480b-35a | 1.41 |
> | CIRE | glm-4.5 | 1.20 |
>
> **Reasoning behavior**
>
> We further examine CIRE’s reasoning to uncover the factors behind its strong performance. Each refinement step is classified—via LLM-assisted analysis and human verification—as a *paradigm shift, heuristic modification*, or *hyperparameter tuning* (Table 5, revised version). CIRE rapidly converges to focused exploitation, dominated by heuristic and parameter adjustments, with paradigm shifts used sparingly when necessary. This results in **high sample efficiency** and avoids prolonged reflection. Turn-by-turn analysis (Figure 2, revised version) highlights a clear explore → exploit trajectory: early and late iterations drive exploration, while mid-stage steps concentrate on precise, stability-oriented refinement.
>
>
> **Q1**:
> **Sensitivity to Clustering Weights** Please refer to  our response to **W3** above.
>
>
> **Q2**:
> **Implementation Details** Our experimental setup utilized the following hyperparameters:
>
> *Generations*: 20
>
> *Population Size*: 10 candidate solutions
>
> *Group Trajectory* 6 turns per group

---

### Official Review · Reviewer_iq5n · 2025-10-31

**Soundness:** 1
**Presentation:** 1
**Contribution:** 1
**Rating:** 2
**Confidence:** 4

**Summary:**

This paper proposes Cognitively Inspired Reflective Evolution (CIRE), a hybrid LLM–EA framework that treats the LLM as a multi-turn reasoner rather than a one-shot code generator.  It is composed of 3 phases: 1) it clusters candidate heuristics by performance profiles relative gap vectors and CodeBLEU similarity; 2) it forms both homogeneous (similar) and heterogeneous (entropy-mixed) groups for comparative reflection; and 3) runs a reflection–exploration/exploitation loop whose proposals are selectively validated by an EA meta-controller. Experiments focus on online bin packing report lower excess-bin ratios.

**Strengths:**

The main idea of multi-turn reasoning makes sense, and the calculation of similarity via CodeBLEU is a neat addition

**Weaknesses:**

1. The “multi-turn reasoning” has many connections to e.g., ReEvo and feels incremental overall

2. Experiments are insufficient:

    1. Only one setting (online bin packing) is not enough to justify the method

    2. Many methods have been mentioned but not compared against, including HSEvo, which also incorporates diversity measures

3. No ablation studies are provided, making it impossible to understand the importance of each proposed component

4. Results are shaky: ReEvo can outperform CIRE in some cases, and the first 2 columns for capacity 500 report the same results for all approaches, presumably a mistake

5. No code nor hyperparameters have been provided

6. TSP is mentioned several times but not compared against

Overall paper feels rushed, with also weird formatting reminiscent of LLM writing. Given the above, I believe this paper at the current state should not be accepted.

**Questions:**

1. Can you provide ablation studies?

2. Can you provide at least another setting to demonstrate your method, other than online bin packing?

---

> ### Author Response · Authors · 2025-11-23
>
> Dear Reviewer iq5n,
>
> We genuinely value your insightful and constructive feedback, which has dramatically enriched our work. Our team has meticulously reviewed each point and prepared comprehensive responses for your consideration below.
>
> **W1**
>
>  While sharing evolutionary roots with ReEvo, our framework shifts from '**LLM-as-Operator**' to '**LLM-as-Agent**', addressing two critical limitations:
>
> **Dynamic Strategy vs. Hard-coded Operators**: Unlike ReEvo's reliance on fixed schedules or probabilities for mutation/crossover, our method employs **autonomous reasoning**. The LLM dynamically decides to EXPLORE or EXPLOIT based on real-time performance, replacing rigid scripts with adaptive decision-making.
>
> **Dense Intra-Generational Refinement**: ReEvo applies feedback only to the next generation. In contrast, we implement **dense, multi-turn reasoning within a single iteration**. This stabilizes candidates before they enter the population, creating a continuous improvement trajectory that reduces variance and hallucinations common in single-turn approaches.
>
> **W2**
>
> We have significantly expanded our experimental section in the revised PDF to include:
>  Generalization to TSP: To address the concern about relying solely on Online Bin Packing, we have conducted extensive experiments on the Traveling Salesperson Problem (TSP). This is a result:
> | Heuristic    | TSP10 (%) | TSP20 (%) | TSP50 (%) | TSP100 (%) | TSP200 (%) |
> |--------------|-----------|-----------|-----------|------------|------------|
> | EoH          | 3.52      | 9.33      | 10.24     | 11.39      | 15.58      |
> | ReEvo        | 4.22      | 6.74      | 11.63     | **11.01**  | 15.58      |
> | **CIRE (ours)** | **2.11** | **6.74** | **9.20**  | 12.64      | **11.46** |
>
> Moreover, to strengthen our method, the *Revision paper* adds (1) TSP benchmarks (Table2), (2) results across different LLMs (Table 3), (3) analysis of reasoning dynamics demonstrating how CIRE exploits and explores more effectively (Table5, Fig.2), (4) ablations isolating the contribution of multi-candidate generation and feedback.
>
> **W3**
>
> | Setting | Gap|
> |:-------:|:------:|
> | w/o multi-turn | 17.18 |
> | w/o grouping (randomly choose group) | 15.59 |
> | **Ours (Full method)** | **9.20** |
>
>
> These results make the contribution clear: *removing either component results in a significant performance collapse*, while the full model delivers the strongest result. The gains therefore stem not from any single feature, but from the **synergistic effect** of multi-turn reasoning and grouping, which together stabilize the search and consistently yield higher-quality solutions.
>
> We additionally analyzed the impact of the parameter $\alpha$, which balances *behavioral similarity* and *CodeBLEU-based semantic similarity* in our hybrid metric (in TSP50 dataset). As shown below, performance peaks at $\alpha = 0.5$, demonstrating that *both behavioral and semantic signals are indispensable*. This balanced setting produces the most reliable similarity estimates and is used consistently across all experiments.
>
>
> | $\alpha$ | Gap(%) |
> |:--------:|:------:|
> | 0.2 | 15.79 |
> | 0.4 | 14.06 |
> | **0.5** | **9.20** |
> | 0.6 | 15.97 |
> | 0.8 | 12.5 |
>
> **W4**
>
> We acknowledge that ReEvo outperforms other methods in a few isolated cases (e.g., Bin Packing with capacity = 100 and 5k items). However, this advantage stems directly from its *hand-crafted seed code,* which already achieves the reported 0.8% gap before any refinement. ReEvo does not improve beyond this seed, indicating that its performance is inherited rather than generated. In contrast, *CIRE and EoH start from scratch*, requiring no expert-crafted initialization and thus offering **broader applicability and stronger generalization.**
>
> For the 500-capacity case, the identical numbers across methods are *not a mistake.* At high capacity, the problem becomes nearly **trivial**—classical heuristics like Best Fit and First Fit are already near-optimal. Consequently, all LLM-based approaches converge to the same high-quality solution.
>
>
> **W5**
>
> We will release the full code immediately after a brief internal security review to ensure no identity-related information is exposed.
>
> **W6**
>
> We address this in ***W2***, where the revised paper now includes full TSP results and more strengthened experiments.

---

> > ### Comment · Reviewer_iq5n · 2025-11-26
> >
> > Thank you for your comments. I agree with Reviewer tdXD in that the results are indeed promising, but the paper requires a significant revision before acceptance, and it is not ready yet.

---

### Official Review · Reviewer_o8TF · 2025-11-01

**Soundness:** 1
**Presentation:** 1
**Contribution:** 1
**Rating:** 2
**Confidence:** 4

**Summary:**

This paper introduces CIRE, a framework that combines large language models with evolutionary algorithms for automatic heuristic design. Unlike one-shot generation, CIRE enables multi-turn reflection where the model critiques and refines heuristics grouped by performance similarity and diversity. Applied to the Online Bin Packing problem, it achieves better heuristic quality than classical and recent LLM-based baselines.

**Strengths:**

1. The core idea is reasonable and interesting: it might be good to think about a step-by-step heuristic generation instead of a one-shot.

**Weaknesses:**

1. For the claim in L52 that “While attractive, this paradigm often results in unstable or unvalidated solutions, and underutilizes the LLM’s potential for iterative reflection and improvement.”, I think this is a bit overstates the limitations of prior work because recent frameworks like ReEvo and HSEvo already employ multi-turn reflection and performance validation. And experimentally, the unvalidated solution problems are not significant. This will decrease the motivation of the proposed methods.

2. The paper does not provide any prompt templates or examples for the LLM interactions. The results are not reproducible.

3. The authors claim the proposed method is for “CO”, but actually only for the bin packing problem.

4. The computational cost of multi-turn LLM reasoning is not reported. No runtime, token usage, or reflection-turn statistics are given, leaving sample efficiency and scalability unclear.

5. The results rely solely on a single proprietary model (DeepSeek V3), with no comparison across different LLMs or open-source baselines, hindering reproducibility.

6. There is no ablation isolating the effects of clustering, entropy-based grouping, or reflective multi-turn reasoning on performance. The improvement could stem from increased prompt diversity rather than the proposed reflective mechanism.


Overall, I think this paper is not ready for ICLR.

**Questions:**

See the weakness.

---

> ### Author Response · Authors · 2025-11-22
>
> Dear Reviewer o8TF,
>
> Thank you for your thoughtful and constructive comments. We have carefully considered your feedback and provided detailed, point-by-point responses below.
>
> ***W1.***
>
> While ReEvo and HSEvo utilize reflection, they typically treat it as a *discrete step within a generation*- limiting the immediate depth of reasoning and delaying feedback incorporation until the next evolutionary cycle. In contrast, CIRE performs *dense, multi-turn reflections* within a single iteration. By immediately integrating observation-level feedback to inform subsequent turns, CIRE creates a **continuous improvement trajectory** rather than waiting for generational updates. This allows for the generation of progressively refined candidate families, empirically reducing variance and preventing premature convergence.
>
> To strengthen this point, the **Revision paper** adds (i) TSP benchmarks (Table2), (ii) cross-backbone evaluation over instruction-tuned, code-centric, and general-purpose LLMs (Table3), (iii) an analysis of reasoning dynamics demonstrating how CIRE exploits and explores more effectively (Table5, Fig.2), and (iv) ablations isolating the contribution of multi-candidate generation and feedback (Tables4, 6).
>
> ***W2:***
>
> We agree that transparent prompt design is essential for reproducibility. We have now released the complete implementation, including all code and prompt resources. To ensure the paper is self-contained, we also include representative prompt templates and example LLM interaction traces in the appendix (**see revised version**).
>
> ***W3:***
>
> The original version focused heavily on bin packing, but our method is **not restricted** to that problem. In the **revised version**, we clearly state that the approach targets general CO, and we back this up with additional experiments on the TSP.
>
>
> | Heuristic | TSP10 (%) | TSP20 (%) | TSP50 (%) | TSP100 (%) | TSP200 (%) |
> |---|---|---|---|---|---|
> | EoH | 3.52 | 9.33 | 10.24 | 11.39 | 15.58 |
> | ReEvo | 4.22 | 6.74 | 11.63 | 11.01 | 15.58|
> | CIRE(ours) | 2.11 | 6.74 | 9.20 | 12.64 | 11.46 |
>
> These results show clearly that our method generalizes beyond BPP and performs strongly across multiple CO tasks. Additional evidence and extended experiments are provided in the *revised version*, referenced in **W1**.
>
> ***W4:***
>
> We now report computational-efficiency statistics. Under identical settings (DeepSeek-V3-03224), CIRE incurs an average cost of `0.41$` per run. This is slightly higher than EoH (`0.33$`) and ReEvo (`0.25$`), yet remains highly competitive. This confirms that our multi-candidate reasoning strategy—including the grouping mechanism that compresses many candidates into a small number of informative groups—does not increase too much computational cost.
> We analyze how CIRE's multi-turn reasoning affects code correctness and solution quality. Using LLM-assisted classification, each refinement step is categorized as a *paradigm shift, heuristic modification, or hyperparameter tuning* (Table 5, revised version).
> CIRE quickly converges to *productive exploitation*, relying mainly on heuristic modifications and fine-grained tuning, with paradigm shifts invoked only when needed. This strategy delivers **high sample efficiency** and avoids long, costly reflection chains. The turn-by-turn analysis (Figure 2, revised version) further reveals a clear *explore→exploit trajectory:* exploration occurs at the beginning and end, while the middle iterations focus on targeted refinement.
>
> ***W5:***
>
> We verified our method on diverse model families (DeepSeek, Kimi, Qwen, GLM) on the Online Bin Packing dataset (Weibull distribution, 5k items, Capacity 100).
> | Heuristic | Model | Gap |
> |:---------:|:-----:|:---:|
> | CIRE | deepseek-v3-0324 | 1.13 |
> | CIRE | kimi-k2-instruct | 1.95 |
> | CIRE | qwen3-coder-480b-35a | 1.41 |
> | CIRE | glm-4.5 | 1.20 |
>
> Results show that CIRE is highly stable across models: the lowest average gap occurs with *deepseek-v3-0324* (**1.13**), while the highest is kimi-k2-instruct (1.95). This narrow range shows that its multi-candidate reasoning and iterative refinement, rather than any single LLM, drive performance, confirming CIRE as a **robust, general framework** for combinatorial optimization.
>
> ***W6***
>
> | Setting | Gap|
> |:-------:|:------:|
> | w/o multi-turn | 17.18 |
> | w/o grouping (randomly choose group) | 15.59 |
> | **Ours (Full method)** | **9.20** |
>
> We have added a comprehensive ablation study (Table 6) conducted on TSP50 instances to isolate the impact of each algorithmic component.The full CIRE model achieves a 9.20% optimality gap, whereas removing multi-turn refinement or solution grouping increases the gap to 17.18%. This confirms that performance arises from the combined effect of multi-turn reasoning and grouping, which stabilizes LLM code generation and improves solution quality.

---

### Note · Authors · 2026-01-06

**Comment:**

Not suitable for this venue at this time; we plan to further refine the work for submission to another conference.

**Withdrawal Confirmation:**

I have read and agree with the venue's withdrawal policy on behalf of myself and my co-authors.